# Innovation in the Development of Synthetic and Natural Ocular Drug Delivery Systems for Eye Diseases Treatment: Focusing on Drug-Loaded Ocular Inserts, Contacts, and Intraocular Lenses

**DOI:** 10.3390/pharmaceutics15020625

**Published:** 2023-02-13

**Authors:** Letizia Pelusi, Domitilla Mandatori, Leonardo Mastropasqua, Luca Agnifili, Marcello Allegretti, Mario Nubile, Assunta Pandolfi

**Affiliations:** 1Department of Medical, Oral and Biotechnological Sciences, Center for Advanced Studies and Technology-CAST, StemTeCh Group, University G. D’Annunzio of Chieti-Pescara, 66100 Chieti, Italy; 2Department of Medicine and Aging Science, Ophthalmology Clinic, University G. D’Annunzio of Chieti-Pescara, 66100 Chieti, Italy; 3Dompé Farmaceutici SpA, Via Campo di Pile, 67100 L’Aquila, Italy

**Keywords:** ocular disease, drug delivery system, ocular insert, contact lenses, intraocular lenses

## Abstract

Nowadays, ocular drug delivery still remains a challenge, since the conventional dosage forms used for anterior and posterior ocular disease treatments, such as topical, systemic, and intraocular administration methods, present important limitations mainly related to the anatomical complexity of the eye. In particular, the blood–ocular barrier along with the corneal barrier, ocular surface, and lacrimal fluid secretion reduce the availability of the administered active compounds and their efficacy. These limitations have increased the need to develop safe and effective ocular delivery systems able to sustain the drug release in the interested ocular segment over time. In the last few years, thanks to the innovations in the materials and technologies employed, different ocular drug delivery systems have been developed. Therefore, this review aims to summarize the synthetic and natural drug-loaded ocular inserts, contacts, and intraocular lenses that have been recently developed, emphasizing the characteristics that make them promising for future ocular clinical applications.

## 1. Introduction

The eye is one of the most important and complex organs in the human body. It can be divided into two principal compartments, namely the anterior and posterior segments, which consist of different layers with specific internal structures. The cornea is located in the outer anterior segment of the eye, followed by the anterior chamber, pupil, iris, and crystalline lens, while the conjunctiva is the thin mucous membrane that lines the inside of the eyelids and covers the sclera [1]. Instead, the vitreous humor, retina, macula, optic nerve, choroid, and sclera are included in the posterior segment. In addition, four main ocular barriers are present, the corneal, tear film, blood–aqueous, and blood–retinal barriers (Figure 1) [2].

Considering the two principal compartments of the eye, the disorders affecting this organ can be distinguished based on the damaged site. Some of them are non-specific, such as infections, traumatic damages, inflammatory diseases, and genetic illnesses, which may affect both the anterior and posterior parts [3]. Some others are mainly related to the limited involvement of specific eye structures; for example, keratoconus for the cornea, diabetic retinopathy (DR) and age-related macular degeneration (AMD) for the retina, and cataracts for the crystalline lens [4].

Therefore, based on the different types of eye disease, many routes of drug administration can be chosen among the conventional ocular dosage forms that are commonly used. These include the topical, systemic (oral or intravenous), periocular (subconjunctival, subtenon, retrobulbar, and peribulbar), and intraocular administration routes [5].

Anterior ocular diseases are usually treated with drugs in the form of topical eye drops that act on the cornea, conjunctiva, sclera, and iris. However, due to the anatomical and physiological barriers of the eye, the bioavailability of the active substance released is less than 5% [6]. Therefore, frequent topical administration is needed, which can induce patient discomfort [7].

Intravenous administration is mainly used to deliver drugs to treat conditions affecting the posterior segment of the eye, even if the presence of both blood–aqueous and blood–retinal barriers still represents an important limit to drug permeation in the eye [8]. Therefore, more invasive approaches, including periocular and intraocular injections, are currently used for the treatment of most posterior ocular diseases [9]. However, to achieve a therapeutic effect, repeated injections are necessary, and this is linked to the development of several adverse effects for the patients, such as inflammation and the risk of infections [10]. For this reason, oral administration is the most widely used and preferred route of administration, since compared to periocular or intraocular injections it represents a non-invasive approach. However, this method of administration also presents an important limitation, since to reach a significant therapeutic efficacy in the target ocular tissues, a high oral dosage is required, which could result in systemic side effects [11]. Therefore, one of the main challenges in ophthalmological research is the development of safe and effective systems able to deliver and release a therapeutic concentration of active drugs directly in situ in the eye [12,13,14].

In recent decades, the important implementation of materials and technologies has allowed the development of different promising ocular drug delivery systems aimed at extending the drug release, bioavailabilty, and pharmacological action in the ocular site, thereby reducing their systemic absorption and side effects and patient discomfort. Concerning the materials employed in the manufacture of these ocular devices, recently the use of both synthetic and natural polymers was authorized by the U.S. Food and Drug Administration (FDA) for their physical and chemical properties and their good biocompatibility profile. These include the synthetic polymers poly-ethylene-glycol diacrylate (PEGDA), polylactic acid (PLA), poly lactic-co-glycolic acid (PLGA), metacrylate (MC), and polyvinyl alcohol (PVA), and natural ones such as hyaluronic acid, chitosan, and gelatin [15,16,17]. 

In addition, new innovative technological approaches able to increase the drug loading and release were developed and approved. These include the use of permeation enhancers for their mucoadhesive properties [18], viscosity modifiers able to improve the drug’s ocular retention [19], as well as nanotechnology approaches based on the use of nanoparticles as carriers that can protect and release active molecules, thereby increasing their ocular residence time [20,21]. Furthermore, recently the thiolated cyclodextrin emerged as a useful excipient for the development of innovative ocular drug delivery systems, since it is able to improve a drug’s ocular residence time and bioavailability thanks to its high mucoadhesive properties and its ability to efficiently solubilize hydrophobic active compounds [22,23].

Therefore, based on the above, this review aimed to report and summarize the innovation in the development of synthetic and natural ocular drug delivery systems, with a focus on the most recently developed drug-loaded ocular inserts (OIs), contact lenses (CLs), and intraocular lenses (IOLs). In detail, particular attention was paid to the fabrication methods used, the drug loading techniques employed, and the potential use in the clinical scenario of such ocular devices. 

## 2. Materials and Methods

The search strategy used in this review was performed employing the electronic database PubMed, applying the following specific keywords: ocular drug delivery, ocular implants, contact lenses, intraocular lenses, sustained release. A filter for years was added, and only original articles published between 2018 and 2022 were considered. 

## 3. Ocular Inserts

### 3.1. Description and Mechanisms

Ocular inserts (OIs) are defined as sterile, thin, multilayered, drug-embedded synthetic devices with a solid or semisolid consistency used to provide medications to the surface of the eye. They are mainly developed to give an adequate alternative solution to the challenging issue of short precorneal medication residence times [24], with the primary objective of extending the drugs’ contact time with the conjunctival tissue [25]. Therefore, OIs can be directly inserted into the conjunctival sac, located between the lower eyelid and the eye. Here, they can ensure the controlled and sustained release of the drugs via three main mechanisms: diffusion, osmosis, or bioerosion (Figure 2). In the diffusion process, the drug is released through the tear fluid in a continuous and controlled manner. Regarding osmosis, it is typical of OIs characterized by the presence of an elastic membrane. In fact, when the device is inserted in the conjunctival sac, the water present in the eye enlarges this elastic membrane, thereby inducing the release of the drug. Finally, bioerosion occurs in the bioerodible OIs and is achieved by the direct erosion of the matrix when the device meets the aqueous microenvironment of the eye [26]. Based on the physical properties, OIs can also categorized as insoluble, soluble, or bioerodible. While insoluble inserts should be removed from the eye once they have released the drug, soluble and bioerodible inserts slowly dissolve over time and do not need to be removed [20]. In addition, concerning the drug loading process, in the insoluble OIs, a drug reservoir is placed between the multilayers and the drug is dispersed or dissolved as a liquid, gel, colloid, semisolid, or solid matrix. Instead, the soluble inserts are loaded with drugs via immersion in the drug solution, and the amount of drug that is loaded depends on several factors, including the presence of a binding agent, the concentration of the drug solution, and the duration of the soaking process. Finally, the bioerodible OIs are characterized using a homogeneous dispersion of the obtained drug thanks to a hydrophobic coating applied to the ocular device’s surface [26].

As previously reported, the main objective of the use of OIs as drug delivery systems is to increase the contact time between the pharmacological molecules and the conjunctival tissue, thereby improving the release and therapeutic efficacy of drugs commonly administered through topical or systemic routes [27]. Indeed, drug-loaded OIs present important advantages compared to the traditional delivery systems, including better drug bioavailability and activity. Furthermore, OIs allow accurate drug dosing since they are loaded with a precise dose of the active compound, which in turn is totally retained at the administration site following its release. Therefore, this last aspect makes it possible to reduce the possible systemic or visual side effects and the number of administration cycles, thereby improving patience compliance [25]. Nevertheless, OIs also presents some disadvantages. These include the difficulties faced in their placement and management. Furthermore, their solid or semisolid consistency can induce irritation and patient discomfort due the perception of an extraneous body in the eye. Moreover, the possible movement of OIs around the eye could interfere with vision [28]. However, the drug-loaded OIs could represent a significant advancement in the therapy of eye disease, and in the last few years several studies have been conducted to evaluate their possible use as ocular drug delivery systems for ocular disease treatment.

### 3.2. Ocular Insert Drug Release Studies

Among the several uses of OIs described in this paragraph (Table 1), it is important to mention the drug delivery of antimicrobial compounds usually used for the treatment of the common ocular infections such as bacterial keratitis [29]. 

In this field, by using the cross-linking approach, innovative OIs composed of the synthetic biopolymer sodium hyaluronate (HA) and the corneal permeation enhancer hydroxypropyl-β-cyclodextrin (HP-β-CD) loaded with cyclosporine (CyS) were developed. The release tests, conducted under a continuous flow that simulated the tear fluid, revealed the controlled release of the antimicrobial compound during the first 1 h. Interestingly, the cyclosporine-loaded OIs characterized by the low HP-β-CD content were able to release the active cyclosporine more quickly. Therefore, the authors demonstrated that by changing the HA/HP-β-CD weight ratio in the OIs, it is possible to regulate the rates of drug swelling and release in ocular tissues. Therefore, the OIs based on the cross-linked HA/HP-β-CD seem to provide an appropriate delivery platform for antimicrobial peptides on the ocular surface [30]. On this basis, the same authors made further ophthalmic OIs based on HA nanofibers loaded both with the antioxidant compound ferulic acid (FA) and the antimicrobial peptide ε-polylysine (ε-PL). Of note, the data demonstrated that the HA-based nanofiber OIs showed an adequate thickness, a good release pattern profile, in vitro biocompatibility, and high antibacterial activity against Pseudomonas aeruginosa and Staphylococcus aureus, suggesting its potential use for the management of different ocular surface infection diseases [41]. 

In addition to these studies, Terreni and colleagues, in an article published in 2020, developed antimicrobial OIs with the aim of finding a good delivery system suitable for application in the precorneal area and able to overcome the issue of the instability of antimicrobial peptide release. In detail, through the freeze-drying technique, the authors produced OIs composed of the synthetic mucoadhesive polymers hydroxypropyl methylcellulose (HPMC) and HA. This insert was loaded with the synthetic antimicrobial peptide derived from the *N*-terminus of human lactoferrin (hLF 1-11) through its direct addition to the polymeric solution before the free-drying. The data demonstrated that this developed OIs can deliver the antimicrobial compound in the precorneal area representing a potential carrier useful for the treatment of infectious phenomena occurring on the ocular surface [31]. 

In this context, new lipophilic sodium-alginate-based OIs were investigated for the sustained release of linezolid (LNZ), a solution usually used for the treatment of antibiotic-resistant Gram-positive bacterial ocular infections, especially induced by methicillin-resistant Staphylococcus aureus [32]. Given that sodium alginate is a good biocompatible carrier but which induces the rapid release of its loaded drug, novel alginate co-polymers were created combining butyl methacrylate (BMC) or lauryl methacrylate (LMC) through the grafting method. The latter is a well-established and powerful technique used for the development of hybrid materials through the co-polymerization of different monomers onto a polymer backbone [42,43]. Regarding the loading approach, LNZ was directly added to the polymeric solution before drying. The results demonstrated that although both co-polymers showed better strength for LZ compared to the alginate polymer alone, the co-polymer derived from the grafting process with alginate and LMC presented the sustained release of LNZ, suggesting its use as a carrier for the development of innovative OIs [32].

Among the innovative techniques employed for the development of new synthetic OIs, electrospinning was largely used, since it is a versatile method that allows to be obtained nanofibers with a large surface area, high flexibility, and better mechanical properties, and with the ability to encapsulate multiple drugs and control their release profile [44]. In detail, electrospinning takes the advantages from electrical forces to obtain nanofibers from polymeric solutions [45].

By using this approach, nanofiber-based OIs were loaded with fluocinolone acetonide (FA) for the treatment of retinal disease [35]. In particular, the authors have chosen the biodegradable polymer polycaprolactone (PCL), which thanks to its biocompatibility, excellent mechanical behavior, and slow degradation time, is largely used for drug delivery, particularly for delivery in the posterior eye segment [46]. Interestingly, the preliminary findings lead the authors to conclude that OIs based on PCL nanofibers could be a promising ophthalmic delivery system for the retinal segment, since they demonstrated that the produced nanofibers were sterile, smooth, and able to extend the drug release behavior, thereby improving the therapeutic amounts of the drug at the target site [35].

An electrospinning technique was also adopted to design polycaprolactone/polyethylene glycol (PCL/PEG) nanofibers containing besifloxacin HCl (BH), a topical chlorofluoroquinolone approved by the FDA for the treatment of bacterial keratitis. Notably, BH was added to the OIs alone or in combination with the corneal permeation enhancer HP-β-C. The drug loading with the permeation enhancer was achieved using the freeze-drying technique. To improve the loading of the HP- β-C/BH complex, the nanofibers were further coated with mucoadhesive polymers such as sodium alginate (SA) or thiolated sodium alginate (TSA). Interestingly, by using a rabbit model of bacterial keratitis, the authors demonstrated that both SA and TSA drug-loaded inserts as well as inserts containing HP-β-C were effective in releasing the BH upon a single-dose application. Therefore, by reducing the drug application frequency, the authors assumed that such novel OIs may be potential substitutes for common therapies used to treat eye infection conditions such as bacterial keratitis [33].

Regarding the employment of OIs for posterior ocular disease treatment, recently Alambiaga-Caravaca and colleagues (2021) aimed to develop, characterize, and evaluate the effect of OIs for the administration of progesterone (PG), a sexual hormone frequently used for its neuroprotection activity in posterior degenerative ocular diseases associated with oxidative stress [47,48,49]. Importantly, since PG is insoluble in water, in this case the permeation enhancer technology was also employed. Therefore, PG was incorporated in β-cyclodextrins able to solubilize the hydrophobic drug. Interestingly, the in vitro studies revealed that the OIs obtained following the grafting of 59% polyvinyl alcohol (PVA), 39% polyvinylpyrrolidone K30 (PVP-K30), and 2% propylene glycol (PGL) was able to sustain the release of PG, which showed controlled diffusion. These data were confirmed by further ex vivo analyses that demonstrated trans-corneal and trans-scleral diffusion of the PG. Therefore, the authors, also considering the good physical and chemical properties of the PG-loaded OIs, claim that its use would be suitable for the treatment of various posterior eye diseases such as DR, AMD, cataracts, glaucoma, and retinitis pigmentosa [36].

The OIs were also combined with nanoparticles (NPs), micelles, vesicles, or microspheres as carriers of active molecules to improve the drug loading, release, and bioavailability [50,51]. 

In this context, employing the spontaneous emulsification technique based on the evaporation of the internal phase of an emulsion through the agitation process [52], OIs composed of hydroxyethyl cellulose (HEC) and polyvinyl alcohol (PVA) were functionalized with a suspension of commercial NPs of Eudragit^®^ L100. In detail, the Eudragit^®^ L100 NPs were loaded with ketorolac tromethamine (KT), an active compound commonly used for the treatment of inflammatory eye disease [53]. As a result, the authors concluded that Eudragit^®^ L100-loaded OIs, thanks to their physical properties such as their flexibility, smoothness, softness, and high release strength, could be efficient ophthalmic drug delivery systems able to prevent frequent drug administration and potentially enhancing patient compliance [37].

In addition, a synthetic OI composed of polyvinylpyrrolidone (PVP) nanofibers was tested to evaluate the release of azithromycin (AZM) incorporated in NPs of poly-lactic-co-glycolic acid (PLGA). Several in vitro, ex vivo, and in vivo analyses were performed by the authors, which demonstrated that the incorporation of drug-loaded PLGA-NPs improved the OIs’ properties and effects. In addition to the increases in drug ocular residence and contact times with conjunctival tissue, more accurate drug dose delivery, sustained and constant drug release, a lower drug administration frequency, increased drug bioavailability, and reductions in the frequency of systemic and visual side effects were observed [34].

In order to provide continuous ocular administration, polycaprolactone (PCL)-based NPs containing atorvastatin calcium (ATC) were investigated as carriers for a sustained anti-inflammatory effect in the eye. In particular, the solvent casting approach was employed. Indeed, through this approach, the polymer is dissolved in an organic solvent and the NPs are added to the solution [54]. Therefore, ATC-laden PCL-NPs were added to a polymeric solution of methylcellulose (MC) and polyvinyl alcohol (PVA). Interestingly, the authors demonstrated that the OIs characterized by the 5% of MC and filled with ATC-laden PCL-NPs showed more sustained in vitro release over the 24 h, suggesting a potentially straightforward, novel, and reasonably priced technique to prepare an ocular insert loaded with promising PCL-NPs able to improve the release and ocular anti-inflammatory activity of atorvastatin [38].

Microspheres of polymethyl methacrylate (PMMA) containing the active molecule ketorolac (KT), a non-steroidal anti-inflammatory drug, were also employed to fill commercial OIs made of coated stainless steel and named ocular coils [39]. Firstly, computed tomography imaging was used to directly examine the shape behavior and tissue contact of the ocular coil in vivo after placing it in a human cadaver’s lower conjunctival fornix. Then, an in vitro lacrimal system was employed to quantify the percentage of ketorolac released. The results revealed that after 28 days, 69.9 ± 5.6% of the loaded ketorolac was delivered from the ocular coil, with a more gradual release after the initial three days [39]. These data were then confirmed using New Zealand White rabbits in which ketorolac released from the coil could be detected in tears for up to 28 days, in aqueous humor for up to 4 days, and in plasma for up to 24 h. In conclusion, according to the promising findings from these two research studies, the tested drug-loaded ocular coil looks to be a useful carrier for the delivery of ophthalmic drugs [40]. 

Instead, polymeric nanomicelles were employed to improve the delivery of cyclosporine A (CyA) to the surface of the eye. [55]. 

Finally, with the aim of enhancing the bioavailability of travoprost (TVP), a synthetic prostaglandin used in glaucoma treatment, spanlastic nanovesicle gels produced with the optimized Box–Behnken method were used [56]. After performing in vivo studies, the authors assessed that the OIs loaded with TVP laden to spanlastic nanovesicles may be a unique ocular delivery strategy for the treatment of glaucoma [57].

## 4. Contact Lenses

### 4.1. Description and Mechanisms

Contact lenses (CLs) are thin, curved plastic lens that are placed on the cornea to correct refractive errors or to protect the eye. Even though the correction of ametropia conditions is their main function, there is an increasing interest in the use of CLs as therapeutic tools to maintain corneal epithelial hydration, alleviate ocular surface pain, promote corneal healing, and provide controlled medications for the treatment of anterior segment ocular disease [58]. Therefore, in recent years, several researchers have focused their studies on possible modifications to apply on commercial or custom-made CLs with the scope for the sustained release of desired drugs in the precorneal area (Figure 3) [59]. This was hypothesized by considering the peculiar structure of the CLs, which are characterized by a three-dimensional polymeric network composed of acrylate or silicone hydrogels that can efficiently absorb and capture drugs when the lenses are soaked into a concentrated drug solution. Therefore, when the obtained drug-loaded CLs are applied to the eye surface, greater drug diffusion toward the corneal surface, more efficient drug penetration, and a lower drug dosage are achieved. As a consequence, a reduction in systemic drug absorption and an improvement in the efficacy of the pharmacological treatments is also obtained [60]. In addition to these advantages, drug-loaded CLs represent a more interesting ocular device compared to the other ones, since they could be used simultaneously to correct ametropia and ocular diseases thanks to the specific drug release [61]. However, CLs present the disadvantage of being released mainly in the anterior segment of the eye. In addition, for these ocular devices, it is necessary to improve the drug loading techniques employed with the goal of increasing the amount of drug that can possibly bind to the CLs. For this reason, as reported in the below paragraph, several approaches were identified and optimized for the development and implementation of innovative drug-loaded CLs.

### 4.2. Contact Lens Drug Release Studies

As summarized in Table 2, several studies were conducted with the aim of investigating the use of CLs as ocular drug delivery systems able to improve the bioavailability of therapeutic agents in the eye.

As previously reported, several researchers have focused their studies on possible modifications to apply to commercial CLs to improve the drug loading efficacy.

A commercial silicon hydrogel CL named ACUVUE^®^ OASYS^®^ was modified to control the delivery of pirfenidone (PFD), an anti-inflammatory and antifibrotic compound used in ocular surgery as a postoperative antiscarring agent [73]. Given that it was previously reported that the addition of vitamin E to silicone hydrogel CLs was able to extend the release of both hydrophobic and hydrophilic active compounds [74,75], in this study [62] an ACUVUE^®^ OASYS^®^ lens was coated with vitamin E particles using the soaking approach, which consists of the direct immersion of the device in a highly concentrated solution, suspension, or emulsion of the drug that can be loaded [58,76]. Interestingly, through in vitro and in vivo analyses, the authors found that the cited commercial CLs modified with vitamin E were more able to sustain the release of anti-inflammatory compounds compared to unloaded vitamin E ones. These promising results allowed the authors to argue that this modification on commercial CLs could be suitable for the delivery of PFD to ameliorate corneal inflammation and fibrosis conditions [62]. Based on this, the same authors soaked with vitamin E the ACUVUE^®^ Oasys^®^ and the 1-DAY ACUVUE^®^ TruEye^TM^ lens to test the release of cysteamine, a compound commonly used for ocular cystinosis treatment [77]. In fact, the oral administration of cysteamine does not provide a therapeutic effect in the cornea, meaning patients require frequent eye drops or face low drug corneal bioavailability [78]. Interestingly, the results confirmed that both Acuvue^®^ Oasys^®^ and ACUVUE^®^ TruEye^TM^ can be efficiently loaded with vitamin E and improved the release behavior of a cysteamine without toxic effects in a rabbit model over a period of 7 days [77]. 

Vitamin E and the commercial ACUVUE Oasys^®^ and ACUVUE TruEye^TM^ lenses were also employed to evaluate the release profiles of ketorolac tromethamine (KT) and flurbiprofen sodium (FS) [63], both non-steroidal anti-inflammatory molecules commonly used in the management of ocular inflammatory conditions. Firstly, the results confirmed the controlled release of both compounds for several days from the vitamin-E-modified CLs. Interestingly, by further coating the vitamin-E-loaded silicone hydrogel lenses with cationic surfactants, chemical substances that carry a positive charge for improved drug loading, the authors achieved a better drug delivery dosage [63]. Based on this, the same authors demonstrated the feasibility of loading and releasing the anti-inflammatory compound diclofenac sodium (DFNa) from hydrogel CLs composed of the synthetic hydrophilic polymer poly-2-hydroxyethyl methacrylate (pHEMA). The pHEMA hydrogel-based CLs were obtained using an oil-in-water microemulsion. The DFNa was, instead, loaded onto the CLs using the soaking method [64].

CLs based on onmethacrylic acid (MAA) were used [66] to improve the loading efficacy of both acyclovir (ACV) and its prodrug valacyclovir (VACV), antiviral compounds employed against ocular keratitis caused by herpes simplex virus (HSV) infection [79,80]. However, the authors demonstrated that only the lenses loaded with VACV are suitable candidates for the preparation of drug-eluting devices useful for treating diseases of the front and back of the eye [66].

Soft hydrogels CLs obtained through the polymerization of a mixture of different monomers including *N*-vinyl pyrrolidone (NVP), 2-hydroxyethyl methacrylate (HEMA) ethylene glycol dimethacrylate (EGDMA), allyl-methacrylate (AMA), e-butyl-hydroxycyclohexyl (TBE) and poloxamer were also employed to achieve the sustained release of naringenin (NAR) [68]. This is a flavonoid compound with important anti-inflammatory and antioxidant properties for treating posterior eye segment diseases. Regarding the loading process, the authors developed two types of NAR-loaded hydrogel lenses: one obtained using a direct drug entrapment method, a technique in which the therapeutic agents are loaded into the device by adding it directly into the prepolymerized monomer mixture before the polymerization process [81], and the other one was the previously mentioned ‘soak and release’ approach [76]. Interestingly, the results highlighted that the CLs prepared using the direct drug entrapment approach provided better-controlled NAR delivery for 24 h, while the ones prepared using the ‘soak and release’ method showed faster NAR release in the first five hours [68].

Ocular infections, including keratitis and conjunctivitis, and related inflammation processes are usually treated with moxifloxacin (MF) and dexamethasone (DM). Therefore, with the goal of overcoming the limits related to the use of the conventional ocular dosage form, chitosan-, glycerol-, and polyethylene glycol (PEG)-based CLs were developed using the solvent casting approach. MF and DM were loaded onto CLs alone or in combination by directly adding the related drug solution to the polymeric formulation. Interestingly, the in vitro and in vivo analyses showed that the mentioned polymeric device was effective in delivering MF and DM at the required therapeutic concentrations, allowing us to hypothesize its employment as a drug-eluting system, especially in postoperative conditions to prevent ocular infections [67].

Among the innovative technologies used for drug delivery, in the last few years particular attention has been given to the use of the three-dimensional (3D) bioprinting for its advantage of producing structures with complex geometries, high levels of accuracy and reproducibility, and controlled drug release profiles [82]. 

Furthermore, Zidan and colleagues (2021) [83] used 3D printing to produce dexamethasone (DM)-loaded CLs composed of both the natural gelatin hydrogels and poly(ethylene glycol) diacrylate (PEGDA). The latter is a synthetic polymer widely used in ocular drug delivery for its ability to prolong the release of loaded therapeutics and its low cytotoxicity against eye cells [84]. 

Additionally, 3D printing technology was also employed to design and develop polylactic acid (PLA)-based CLs for the management of glaucoma through the delivery of timolol maleate (TM) [70]. 

In the field of glaucoma treatment, silicone hydrogel soft CLs soaked with brinzolamide (BRZ) were developed [71]. BRZ is a carbonic anhydrase inhibitor that is effective in reducing intraocular pression (IOP) via topical administration [85]. In particular, the authors demonstrated that the drug loading did not induce significant alterations to the physical properties of the CLs, concluding that this delivery system, thanks also to the constant release of the BRZ, may be a good alternative to eye drops for treating glaucoma [71]. 

Regarding posterior ocular diseases such as DR, which are usually treated via systemic, oral, or intraocular drug administration, silicone hydrogel CLs were employed for the delivery in situ of epalrestat (EPS), an aldose reductase inhibitor used for the treatment of diabetic neuropathy [86]. To synthesize the CLs, the mixture of 2-hydroxyethyl methacrylate (HEMA) and monomethacryloxypropyl-sym-polydimethylsiloxane hydroxypropyl-terminated (MCS-MC12) monomers was subjected to polymerization. Then, the CLs were soaked in the solution of EPS to promote the drug loading. Notably, in vitro and ex vivo experiments were performed, and the results showed that these drug loaded hydrogel CLs may be useful as ocular devices to regulate the posterior release of epalrestat, facilitating its accumulation and diffusion across the cornea [69]. 

As for the OIs, several studies have investigated the feasibility of engineering CLs with micelles and NPs as nanocarriers able to improve drugs delivery [87].

In this context, with the objective of treating dry eye syndrome and overcoming the limit of the drug’s burst release, CLs functionalized with micelles of cholesterol-hyaluronate (C-HA) containing cyclosporine (CyS) were used. In detail, the CLs were produced using the photopolymerization technique, in which visible or ultraviolet light is used to initiate and propagate a polymerization reaction to form a linear or crosslinked polymer structure [88]. In particular, the synthetic polymers hydroxyethyl methacrylate (HEMA) and ethylene glycol dimethacrylate (EGDMA) were employed. After performing both physical and chemical analyses, the authors concluded that the lenses engineered with the C-HA micelles showed improved wettability and mechanical strength. In addition, further in vitro analyses revealed the sustained release of the CyS for up to 12 days. Finally, the therapeutic effects of such C-HA-CyS-CLs was proven in a rabbit disease model of dry eye syndrome [72].

Similarly, CLs engineered with the anti-inflammatory levofloxacin (LEV) previously included in poly-sulfobetaine-methacrylate (PSBMA)-based nanogels were developed [65]. Nanogels are hydrogel NPs with sizes ranging between 1 and 1000 nm that are created using physical or chemical cross-linking and have gained a lot of attention due to their high incorporation efficiency and sustained drug release [89]. CLs instead were made using the two monomers hydroxyethyl methacrylate (HEMA) and *N*-vinyl-2-pyrrolidone (NVP), which were subjected to a polymerization process using the cast molding method [90]. In detail, the authors developed CLs based on various nanogel loading percentages and analyzed their structure, surface morphology, transmittance, mechanical properties, in vitro drug release, and biocompatibility. These analyses revealed that the CLs with an 8 wt% nanogel loading content were able to stably release the LEV for ten days, suggesting its use as an alternative to conventional eye drops or ointment formulations for the long-term treatment of oculopathy [65]. 

Recently, NPs of poly-ethyl-glycole (PEG) were used to engineer CLs with the aim of delivering latanoprost (LTP) in the posterior ocular segment and reducing IOP in glaucoma. The results demonstrated that CLs loaded with PEG-NPs compared to the non-loaded and traditional drug-impregnated lenses have a higher drug absorption capacity and a controlled release rate up to 120 and 96 h. Therefore, in conclusion, the authors assumed that these systems could represent excellent substitutes to the conventional drug delivery systems [91].

Finally, Xu in 2022 designed novel silicone CLs loaded with silica-based NPs containing brimonidine (Bri) to avoid the problem of its high burst release, which occurs using eye drop solutions. Three methods were used to engineer the CLs with Bri-laden silica NPs: the soaking method, direct addition, and microemulsion. Overall, regardless of the loading method that was employed, the in vitro and in vivo experiments demonstrated that silica NPs could be effective in sustaining the release of Bri. In addition, histological tests of the cornea proved that these developed drug-loaded lenses were safe and biocompatible, suggesting their possible future human use [92].

## 5. Intraocular Lenses (IOLs)

### 5.1. Description and Mechanisms

Intraocular lenses (IOLs) are tiny artificial devices placed inside the eye, which have the main function of restoring the refractive power of the natural crystalline lens that is removed during cataract surgery [93]. Most of the IOLs are made of synthetic polymers, which can be divided into two major groups: acrylic and silicone [94]. As a result, unlike the CLs described above, the IOLs are permanent, meaning they are considered devices with intermediate characteristics between OIs and CLs [95]. Given that the IOL is implanted during cataract surgery and remains in the eye after surgery, recently this ocular device has received growing attention for its possible use as an optimal delivery system for intraocular drug release (Figure 4). In particular, the use of IOLs was hypothesized for the treatment of the most common complications occurring following cataract surgery, which include inflammation, infection, and posterior capsule opacification.

Regarding the drug loading process, two possible approaches are described in the literature. The first one is represented by the drug coating on the IOLs’ surfaces through a soaking method [96]. The second one is based on the use of a separate polymeric drug reservoir attached to the IOLs. During the development of drug-loaded IOLs, in addition to the loading method that is adopted, it is necessary to consider further key aspects to avoid the occurrence of important inconveniences. Given that compared to the other ocular systems, IOLs can efficiently release higher amounts of drugs in the intraocular site, they must be loaded with an effective but non-toxic level of the active compound. Therefore, it is necessary to determine the optimal amount of the drug to load and to understand its pharmacokinetics. In addition, the drug loading should not affect the optical properties or the dioptric power of the IOLs and its position when implanted in the crystalline lens to avoid the phenomenon of blurring vision [97]. 

Considering all of these aspects, it possible to develop ideal IOLs able to release desired drugs in the intraocular compartments through the diffusion process. 

Finally, regarding the main advantages of drug-loaded IOLs, compared to the other drug delivery systems usually used to treat cataract postoperative complications, the choice of these ocular devices could lead to better patient compliance and management. Indeed, drug-loaded IOLs could represent permanent drug delivery systems implanted in a single surgical procedure following cataract surgery [98].

### 5.2. Intraocular Lens Drug Release Studies

Based on the characteristic and the advantages described above, recently several studies have paid attention to the development of IOLs loaded with specific drugs (Table 3).

The drugs mainly chosen for the loading studies are antibiotics and anti-inflammatory drugs, which are usually used to control infection and inflammation conditions occurring after cataract surgery. 

In this context, the feasibility of using acrylic IOLs for the sustained release of the antibiotic moxifloxacin (MXF), which is commonly used for endophthalmitis prophylaxis after cataract surgery, was investigated. The acrylic IOLs were obtained via the cross-linking of the synthetic co-polymers 2-hydroxylethyl methacrylate (HEMA) and methyl methacrylate (MMA), while the drug loading was performed by soaking the IOLs in a MXF solution. Interestingly, the drug release results obtained in vitro and in vivo showed that the loaded IOLs allowed the constant release of active MXF for up to 2 weeks [99]. 

Acrylic IOLs were also employed to study the delivery of methotrexate (MTX) [100], an FDA-approved folic acid antagonist [105], to lessen the posterior capsule opacification. Interestingly, the modern technique known as supercritical impregnation [106] was used to load MTX onto the IOLs, and through the use of ex vivo implants in human donor capsular bags, the authors found that the loaded IOLs sustained the release of MTX for more than 80 days, which induced a decrease in fibrosis by preventing the epithelial–mesenchymal transformation of lens epithelial cells [100]. 

Xiang (2020) instead demonstrated the capability of IOLs based on polymer hydrogel to load and delivery indomethacin (IND), a non-steroidal anti-inflammatory compound used to prevent ocular inflammation and posterior capsule opacification [104]. The hydrogel lenses were developed through the free-radical polymerization of 2-hydroxyethyl methacrylate (HEMA), methyl methacrylate (MMA), and methacrylic acid (MAA). Instead, IND prodrugs were prepared via the esterification of IND and HEMA and then directly added to the polymeric solution before the free-radical polymerization.

Similarly, hydrogel-based IOLs, composed of the polymers HEMA and 2-butoxyethyl methacrylate (BEM), were used [107] to co-deliver steroidal (dexamethasone sodium phosphate, DSP) and non-steroidal (bromfenac sodium, BFS) active compounds for the treatment of pseudophakic cystoid macular edema [108]. Following the drugs’ binding using two positive charge monomers such as *N*-2-aminopropyl-methacrylamid (APMA) and acrylamide (AAm), the results obtained in vivo showed that the drug-loaded IOLs allowed the release of BFS and DSP, which both reached therapeutic concentrations in the aqueous humor for about 2 and 8 weeks, respectively [107]. 

With the aim of improving the drug loading process, several approaches are used to modify the IOLs’ surfaces. Among these, layer-by-layer (LbL) deposition, molecular imprinting, and the coating and the loading of NPs are the most used methods [109].

LbL deposition based on the natural polymers hyaluronic acid (HA) and chitosan (CHI) was used to chemically load the antiproliferative drug paclitaxel (Pac) to prevent posterior capsule opacification following cataract surgery. Importantly, studies in vitro performed to evaluate the drug release highlighted that the HA/CHI multilayer IOLs showed a sustained release profile of Pac, thereby providing support for this novel approach to prevent or treat posterior capsule opacification [101].

Commercial acrylic G-free^®^-based IOLs were tested to load moxifloxacin (MXF) and the anti-inflammatory diclofenac (DFN) for posterior ocular opacification management. In detail, by using the molecular imprinting approach, which creates molecularly imprinted polymers with tailor-made binding sites complementary to the molecules in terms of their shape, size, and functional groups [110], the surfaces of the IOLs were modified with the functional monomers acrylic acid (AA), methacrylic acid (MAA), and 4-vinylpiridine (4-VP) [102]. 

The coating technique was instead employed to modify the IOL surfaces with hydrophilic polydopamine (PDA) via dopamine self-polymerization, a technique that exploits the oxidation of dopamine at alkaline pH using dissolved oxygen [111]. The IOLs were then loaded with the antiproliferative drug doxorubicin (DOX). Interestingly, in vitro and in vivo studies demonstrated that such modified IOLs were safe, biocompatible, and effective in inducing cell apoptosis, assuming their use was to prevent postoperative complications such as posterior capsule opacification [112].

Regarding the use of NPs in combination with IOLs, only one study employing this approach was published in 2013. In particular, the authors modified the surfaces of the synthetic commercial poly-methyl-methacrylate (PMMA) IOLs with chitosan nanoparticles to release 5-fluorouracil (5-Fu), an active compound for the prevention of posterior capsule opacification. The in vitro drug release tests showed the burst release of the 5-Fu from the modified IOLs in the first 2 h, which was sustained for at least 4 days. In addition, the in vivo results performed with New Zealand rabbits demonstrated that at 4 weeks implantation of such nanoparticle-modified IOLs, the animals showed lighter posterior capsule opacification than the control group [103].

## 6. Conclusions and Future Perspectives

Here, we have provided an overview of the most recent and innovative ocular drug delivery systems developed with the aim of overcoming the limits associated with the conventional dosage forms used for drug release in the eyes, such as topical, systemic, periocular, and intraocular administration methods. In addition, particular attention was paid to the description of the materials and techniques employed, the drug-loaded approach that was used, and the potential clinical application for the treatment of both anterior and posterior ocular diseases.

In more detail, the review was focused on the description of three specific ocular drug delivery systems, namely OIs, CLs, and IOLs, as compared to the other traditional drug delivery systems they present important advantages, including their ability to overcome the limits due ocular barriers, their ability for extended and continuous drug release, their better drug bioavailability, and their ability for accurate drug dosing, thereby reducing visual or system side effects and patient discomfort [5]. 

Overall, the studies that have been reported have shown that all synthetic and natural eye devices developed in the last five years showed promising results in terms of safety and efficacy in drug release. However, one of the main challenges of ongoing research in the ophthalmic field is evaluating the feasibility of using, as an ocular drug delivery method, directly native ocular tissues for bioengineering with active compounds. Indeed, although the ocular devices developed so far have been composed of synthetic or natural polymers with good biocompatibility profiles and have been approved by the FDA, the possible employment of the derived ocular tissue with a preserved extracellular matrix (ECM) organization could overcome the important disadvantages associated with the described ocular system. These include irritation, blurred vision, and hypoxia, which can frequently occur following the implantation of artificial devices. In addition, their use could induce patient discomfort due to the perception of an extraneous body in the eye.

In this regard, we recently demonstrated the feasibility of using corneal stromal lenticule tissue as a natural, more biocompatible, and non-immunogenic ocular drug device [113]. In more detail, corneal stomal lenticules are neatly cut discs of native, well-organized, collagen-rich ECMs that are ultrathin (about 30–140-μm-thick), transparent, avascular, and mechanically strong, obtained from young, healthy corneas following the small incision lenticule extraction (SMILE) surgical procedure to correct myopia defects [114]. Their disposal is an enormous waste of a valuable resource of tissue that can be reused for therapeutic uses, especially with the constantly growing number of SMILE surgeries. Therefore, we hypothesized their use as natural ocular drug delivery systems. In particular, in this study, we bioengineered, for the first time, human corneal stroma lenticules directly obtained from patients undergoing SMILE with PLGA microparticles laden with recombinant human nerve growth factor (rhNGF-PLGA), a neurotrophic factor that showed promising therapeutic results for ocular disease treatment [115,116]. Interestingly, through in vitro studies, we demonstrated that the rhNGF-PLGA-engineered lenticules were able to sustain the constant release of active rhNGF for up 1 month, assuming that this native ocular tissue with a preserved ECM and directly obtained from the cornea could be a suitable natural drug ocular device for reimplantation in vivo for drug release in the eye [113].

In conclusion, although the in vitro, ex vivo, and in vivo results reported here are promising, further studies are needed to approve the use of the described artificial and natural ocular drug delivery systems for potential use in eye disease treatment. 

## Figures and Tables

**Figure 1 pharmaceutics-15-00625-f001:**
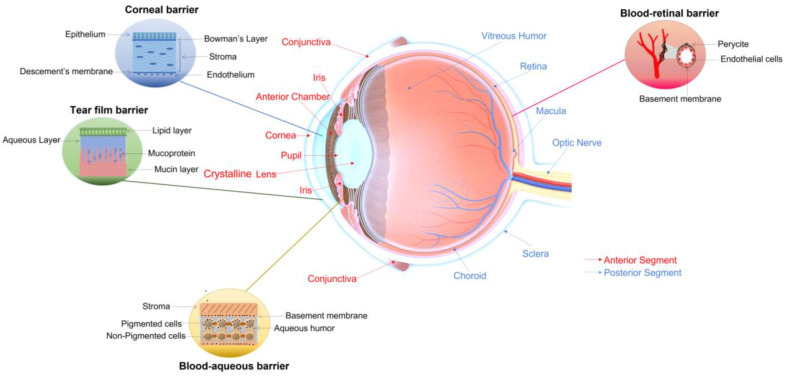
Schematic representation of the eye anatomy. In red are indicated the specific structures located in outer anterior segment, in blue the components of the posterior compartment, and in black the main ocular barriers present in the eye.

**Figure 2 pharmaceutics-15-00625-f002:**
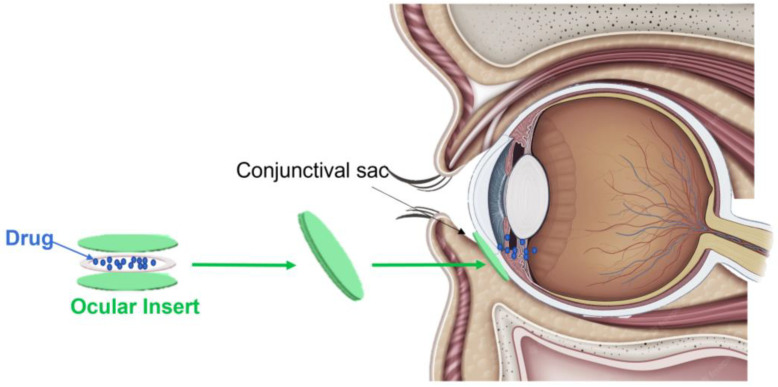
Representative image of a drug-loaded ocular insert (OI). A drug-loaded OI is a thin synthetic device composed of polymeric multilayers loaded with desired drugs. It is directly inserted into the conjunctival sac, successively releasing in a sustained manner the active compound in the anterior segment of the eye via diffusion, osmosis, or bioerosion mechanisms based on its physical properties.

**Figure 3 pharmaceutics-15-00625-f003:**
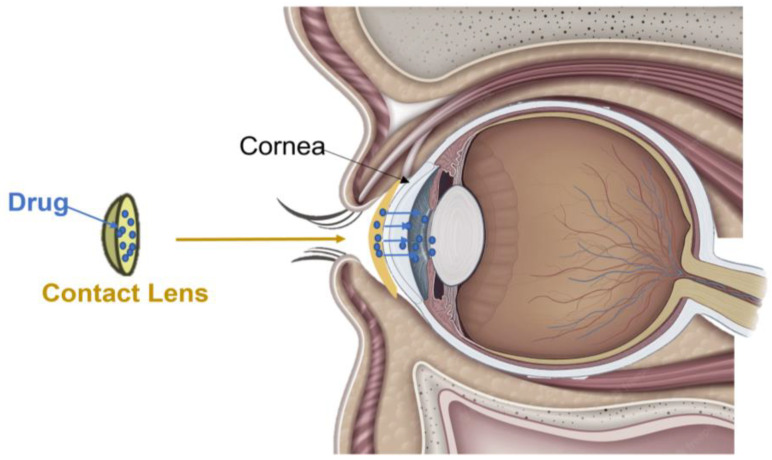
Representative image of a drug-loaded contact lens (CL). A drug-loaded CL is a thin and curved plastic lens characterized by a three-dimensional polymeric network of acrylate or silicone hydrogel able to capture drugs in its polymeric mesh. Once applied to the precorneal area of the eye, the CL can release the loaded drug, mainly in the anterior segment of the eye, through a diffusion mechanism.

**Figure 4 pharmaceutics-15-00625-f004:**
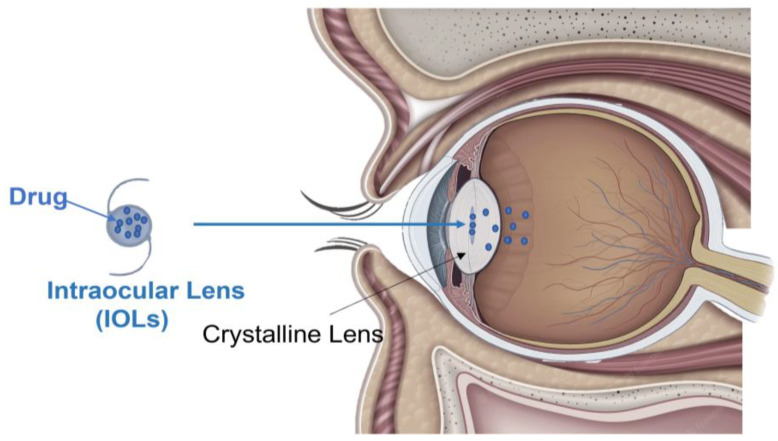
Representative image of a drug-loaded intraocular lens (IOL). A drug-loaded IOL is a permanent, tiny synthetic ocular device composed of acrylic and silicone. Once implanted during cataract surgery into the crystalline lens, it allows sustained intraocular drug release through the diffusion process.

**Table 1 pharmaceutics-15-00625-t001:** Summary of the studies on drug-1oaded OIs.

Clinical Application	OIs Materials	OIs Fabrication Methods	Drug-Loaded	Drug-Loading Technique	Reference
Ocular Infections	Sodium Hyaluronate (HA) combined with Hydroxypropyl-β-Cyclodextrin (Permeation Enhancer)	Cross-linking	Cyclosporine (CyS)	Soaking	[30]
	Hydroxypropyl Methylcellulose (HPMC) combined with Hyaluronate (HA)	Freeze-Drying Technique	N-Terminus of Human Lactoferrin (hLF 1-11)	Directly addition to polymeric solution	[31]
	Alginate combined with Butyl Methacrylate (BMC) Or Lauryl Methacrylate (LMC)	Grafting Method	Linezolid (LNZ)	Directly addition to polymeric solution	[32]
	Polycaprolactone (PCL)/Polyethylene Glycol (PEG) combined with Sodium Alginate (SA) Or Thiolated Sodium Alginate (TSA)	Electrospinning	Besifloxacin Hcl (BH)	Permeation Enhancer Hydroxypropyl- β -Cyclodextrin	[33]
	Polyvinylpyrrolidone (PVP) nanofibers	Electrospinning	Azithromycin (AZM)	Poly-lactic-co-glycolic acid (PLGA) Nanoparticles	[34]
Posterior Ocular Diseases	Polycaprolactone (PCL) nanofibers	Electrospinning	Fluocinolone Acetonide (FA)	Directly addition to polymeric solution	[35]
	Polyvinyl Alcohol (PVA) combined with Polyvinylpyrrolidone K30 (PVP-K30) and Propylene Glycol (PGL)	Grafting Method	Progesterone (PG)	Permeation Enhancer β -Cyclodextrin	[36]
	Hydroxyethyl Cellulose (HEC) and Polyvinyl Alcohol (PVA)	Spontaneous emulsification	Ketorolac Tromethamine (KT)	Eudragit® L100 commercial Nanoparticles	[37]
Ocular Inflammatory conditions	Methylcellulose (MC) and Polyvinyl Alcohol (PVA)	Solvent casting	Atorvastatin Calcium (ATC)	Polycaprolactone (PCL) Nanoparticles	[38]
	Coated stainless steel	Commercial	Ketorolac Tromethamine (KT)	Polymethyl methacrylate (PMMA) Microspheres	[39,40]

**Table 2 pharmaceutics-15-00625-t002:** Summary of the studies for drug-loaded CLs.

Clinical Application	OIs Materials	OIs Fabrication Methods	Drug-Loaded	Drug-Loading Technique	Reference
Ocular Inflammatory conditions	Silicon Hydrogel	Commercial (ACUVUE® OASYS® and ACUVUE TruEyeTM)	Pirfenidone (PFD), Ketorolac Tromethamine (KT) and Flurbiprofen Sodium (FS)	Vitamin E coating	[62,63]
	Poly-2-Hydroxyethyl Methacrylate(pHEMA)	Microemulsion	Diclofenac sodium (DFNa)	Soaking	[64]
	2-HydroxyethylMethacrylate (HEMA) nd N-Vinyl-2-Pyrrolidone (NVP)	Solvent Casting	Levofloxacin (LEV)	Poly (sulfobetaine methacrylate) (PSBMA) based Nanogel	[65]
Ocular Infections	Hydrogel: Methacrylic acid (MAA)	Polymerization of monomers mixture	Acyclovir (ACV) and Valacyclovir (VACV)	Soaking	[66]
	Chitosan, Glycerol and Polyethylene Glycol (PEG)	Cast molding method	Moxifloxacin (MF) and dexamethasone (DM)	Drugs solution added to polymeric formulation	[67]
Posterior Ocular Diseases	Hydrogel: N-vinyl pyrrolidone (NVP), 2-hydroxyethyl methacrylate (HEMA) ethylene glycol dimethacrylate (EGDMA), allyl methacrylate (AMA), e-Butyl-hydroxycyclohexyl (TBE) and Poloxamer	Polymerization monomers mixture	Naringenin (NAR)	Direct Drug Entrapment and Soaking	[68]
	Hydrogel: 2-Hydroxyethyl Methacrylate (HEMA) and Monomethacryloxypropyl-Sym-Polydimethylsiloxane Hydroxypropyl terminated (MCS-MC12)	Polymerization monomers mixture	Epalrestat (EPS)	Soaking	[69]
Glaucoma	Polylactic Acid (PLA)	3D printing	Timolol Maleate (TM)	Drugs solution added to polymeric formulation	[70]
	Silicone	Polymerizaztion of monomers mixture	Brinzolamide (BRZ)	Soaking	[71]
Dry Eye Syndrome	Hydroxyethyl Methacrylate (HEMA) and Ethylene Glycol Dimethacrylate (EGDMA)	Photopolymerization	Cyclosporine (CyS)	Cholesterol-Hyaluronate (C-HA) Micelles	[72]
	Hydrogel: Poly-2-Hydroxyethyl Methacrylate(pHEMA)	Microemulsion	Diclofenac sodium (DFNa)	Soaking	[64]

**Table 3 pharmaceutics-15-00625-t003:** Summary of the studies for drug-loaded CLs.

Clinical Application	OIs Materials	OIs Fabrication Methods	Drug-Loaded	Drug-Loading Technique	Reference
Ocular Infections	2-Hydroxylethyl methacrylate (HEMA) and Methyl Methacrylate (MMA)	Cross-linking	Moxifloxacin (MXF)	Soaking	[99]
	Acrylic	Commercial	Methotrexate (MTX)	Supercritical impregnation technology	[100]
Posterior Capsule Opacification	Hyaluronic Acid (HA) and Chitosan (CHI)	Layer by Layer (LbL)	Paclitaxel (Pac)	Chemical Bonding	[101]
	Acrylic G-free® material [ethylene glycol phenyl ether acrylate (EGPE), 2-hydroxyethyl methacrylate (HEMA) and poly (propylene glycol) dimethacrylate (PPGDMA)]	Commercial	Moxifloxacin (MXF) and Diclofenac (DFN)	Molecular Imprinting	[102]
	Poly-Methyl-Methacrylate (PMMA)	Commercial	5-fluorouracil (5-Fu)	Chitosan Nanoparticles	[103]
Ocular Inflammatory conditions	Hydrogel: 2-hydroxyethyl methacrylate (HEMA), Methyl Methacrylate (MMA), Methacrylic acid (MAA)	Free-radical polymerization	Indomethacin (IND)	Directly addition to polymeric solution	[104]

## Data Availability

Not applicable.

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
