# Peer review of "Innovation in the Development of Synthetic and Natural Ocular Drug Delivery Systems for Eye Diseases Treatment: Focusing on Drug-Loaded Ocular Inserts, Contacts, and Intraocular Lenses"

_pharmaceutics, 2023, doi:10.3390/pharmaceutics15020625_

Round 1

Reviewer 1 Report

The manuscript provides an overview of the recent innovations in ocular drug delivery systems, specifically focusing on ocular inserts, contact lenses, and intraocular lenses. The authors highlight the limitations of current ocular therapies and the need for more effective delivery systems. The manuscript also discusses the use of various techniques and materials in the development of these new drug delivery systems, including both synthetic and natural polymers.

The manuscript is well-organized and clearly written. The introduction and conclusion effectively summarize the main points of the review. However, the manuscript could benefit from some additional information and improvements:

· The manuscript could benefit from a more detailed description of the specific drug delivery systems discussed in the review. This would provide a clearer understanding of the innovations and advancements in this field.

· A more detailed discussion of the challenges facing the development and implementation of these new drug delivery systems should be included. This would provide a complete understanding of the current state of the field.

· Also provide a more detailed discussion of the materials and techniques used in the development of these drug delivery systems.

·   Include ongoing research with synthetic and natural drugs in the field of ocular drug delivery systems in tabular form. This would provide a complete understanding of the future perspectives of the field.

· All the abbreviations should be rechecked, some of them are not defined anywhere in manuscript.

·  The manuscript should also add references for the studies that are already cited in the manuscript and provide the sources for the facts and figures provided in the manuscript.

· The manuscript should also be proofread for any grammatical or typographical errors.

·   Overall, the manuscript provides a good overview of recent innovations in ocular drug delivery systems, but it could benefit from more detailed information and additional discussion of the challenges and future perspectives in the field.

Author Response

Reviewer 1

The manuscript provides an overview of the recent innovations in ocular drug delivery systems, specifically focusing on ocular inserts, contact lenses, and intraocular lenses. The authors highlight the limitations of current ocular therapies and the need for more effective delivery systems. The manuscript also discusses the use of various techniques and materials in the development of these new drug delivery systems, including both synthetic and natural polymers.

The manuscript is well-organized and clearly written. The introduction and conclusion effectively summarize the main points of the review. However, the manuscript could benefit from some additional information and improvements:

We thank Reviewer #1 for the appreciation of our manuscript as well as for his/her useful comments in response to which changes have been made. Particularly, the sections modified or added in the text are highlighted in red as track changes mode.

- The manuscript could benefit from a more detailed description of the specific drug delivery systems discussed in the review. This would provide a clearer understanding of the innovations and advancements in this field.

We thank the Reviewer for his/her observation. Therefore, in the revised version we have added, in separate specific paragraphs entitled “Description and mechanisms”, a more detailed description of the ocular drug delivery systems discussed in this review. Please see the paragraph 3.1 for ocular inserts (page 4), 4.1 for contact lens (page 11) and 5.1 for intraocular lens (page 16)

- A more detailed discussion of the challenges facing the development and implementation of these new drug delivery systems should be included. This would provide a complete understanding of the current state of the field.

This review reported the more recently ocular drug delivery systems developed in the last five years (from 2018 to 2022) giving particular attention to the description of the materials and technologies used for their development and drug loading. As regard the materials employed in the manufacture of these ocular devices, recently the use of both synthetic and natural polymers was authorized by the U.S. Food and Drugs Administration for their physical/chemical properties and good biocompatibility profile. These include PEGDA, PLA, PLGA, MC and PVA, hyaluronic acid, chitosan and gelatin. While, among the most innovative technologies recently developed and approved we reported the use of permeation enhancers for their mucoadhesive properties, viscosity modifiers able to improve drugs’ ocular retention as well as of nanotechnology approaches based on the use of nanoparticles as drugs carrier. As properly suggested by the Reviewer we have better detailed these aspects in the Introduction section (see page 4 lines 98-111).

However, the Reviewer’s comment gave us also the opportunity to better discuss another important challenge in this field. This is represented by the feasibility to use, as alternative to artificial ocular devices, directly native ocular tissues to bioengineer with active compounds. Indeed, the use of a derived-ocular tissue with a preserved extracellular matrix (ECM) organization could overcome important limits due to the use of artificial devices such as irritation, blurring of vision and patient’s discomfort. In this context, for the first time, we have recently demonstrated the feasibility to use corneal stromal lenticule, a discarded tissue directly obtained from cornea during refractive surgery, as natural, biocompatible and non-immunogenic ocular drug device. In detail, we found that corneal stromal lenticule can load and release active recombinant human Nerve Growth Factor (rhNGF) previously laden to PLGA nanoparticles. This allowed us to assume that such native ocular tissue with a preserved ECM could be a suitable natural drug ocular device to reimplant in vivo for the drug release in the eye. Based on this, for the reviewer only, we report that we are performing a pre-clinical study using the New Zealand Rabbit animal model. Of note, the preliminary results obtained show that corneal stromal lenticula loaded with rhNGF-PLGA is effective in the release of the active neurotrophic factor also in vivo. Please see the” Conclusion and Future Perspectives” paragraph pages 21-22 lines 751-773.

- Also provide a more detailed discussion of the materials and techniques used in the development of these drug delivery systems

In the revised version we have better detailed and discussed the materials and techniques used in the development of the three ocular drug delivery systems.

- Include ongoing research with synthetic and natural drugs in the field of ocular drug delivery systems in tabular form. This would provide a complete understanding of the future perspectives of the field.

As properly suggested by the Reviewer, to make our review more attractive for the readers, we have added three tables in which, for each ocular devices described, we have summarized the studies discussed in this review (see pages 7, 13 and 19).

- All the abbreviations should be rechecked, some of them are not defined anywhere in manuscript.

We rechecked and defined all abbreviations in the text.

- The manuscript should also add references for the studies that are already cited in the manuscript and provide the sources for the facts and figures provided in the manuscript.

As suggested by the Reviewer, we have added in the text further references which are highlighted in red in the reference paragraph (see page 23). As regard the figures sources, all figures reported in the review are original and were totally realized by the authors. Therefore, they do not require the addition of the source.

- The manuscript should also be proofread for any grammatical or typographical errors.

We are sorry for grammatical or typographical errors. Therefore, we have rechecked and corrected all of them.

Reviewer 2 Report

·         This review is titled ocular drug delivery system but focused only on drug loaded-ocular inserts, contact, and -intraocular lens. So, it’s better to revise the title, as drug delivery system is a very vast term and in this review, the whole area of DDS is not covered.

·         The summarized presentation of data in a review article is the most attractive way for readers. Unfortunately, data is not presented in a summarized form (no table or influential figure is present in the review article).

·         Mechanism of these ocular inserts, contact, and -intraocular lens should be discussed in a separate section. Moreover, the drug release patterns should be elaborated.

·         What is the advantage of ocular inserts, contact, and -intraocular lens over other DDS, should be addressed.

·         Demerits (if any) of ocular inserts, contact, and -intraocular lens should be highlighted

·         A figure should be self-explanatory, unfortunately only one figure is used for ocular inserts, contact, and -intraocular lens with no explanation at all.

·         Introduction section must be improved. No detailed information on other conventional dosage forms used for ocular delivery is added. For instance, thiolated CDs are used as ocular permeation enhancers due to their better mucoadhesive properties.

·         Conclusion is usually a summary of the abstract that highlight the major outcomes. Therefore, the conclusion should be revised in a comprehensive form.

Author Response

Reviewer 2

We thank reviewer #2 for his/her useful comments in response to which changes have been made. Particularly, the sections modified or added in the text are highlighted in red as track changes mode.

-This review is titled ocular drug delivery system but focused only on drug loaded-ocular inserts, contact, and -intraocular lens. So, it’s better to revise the title, as drug delivery system is a very vast term and, in this review, the whole area of DDS is not covered.

We agree with the Reviewer’s comment and based on his/her suggestion, we have revised the title of the review. The new one is the follow: “Innovation in the development of synthetic and natural ocular drug delivery system for eye diseases treatment: focus on drug-loaded ocular inserts, -contact and -intraocular lens”.

- The summarized presentation of data in a review article is the most attractive way for readers. Unfortunately, data is not presented in a summarized form (no table or influential figure is present in the review article).

As properly suggested by the Reviewer, to make our review more attractive for the readers, we have added three tables in which, for each ocular devices described, we have summarized the studies discussed in this review (see pages 7, 13 and 19). In addition, below we report the revised graphical abstract to be submitted as a graphical summary of the review.

- Mechanism of these ocular inserts, contact, and -intraocular lens should be discussed in a separate section. Moreover, the drug release patterns should be elaborated.

We thank the Reviewer for his/her observation. Therefore, in the revised version we have reported, in a separate section entitled “Description and Mechanisms”, the mechanisms of the three drug delivery systems discussed in the review. Please see the paragraphs 3.1 for ocular inserts (page 4), 4.1 for contact lens (page 11) and 5.1 for intraocular lens (page 16).

- What is the advantage of ocular inserts, contact, and -intraocular lens over other DDS, should be addressed.

As suggested by the Reviewer, in the revised version we have reported the advantages and demerits of each ocular device described. Please see page 5 (lines 163-179) for the ocular inserts; pages (lines 383-398) for contact lens; page 17 (lines 595-608). In addition, the advantages and demerits of ocular inserts, contact, and -intraocular lens over other drug delivery systems, are resumed in the “Conclusion and Future Perspectives” paragraph (see page 21 lines 715-732).

- Demerits (if any) of ocular inserts, contact, and -intraocular lens should be highlighted.

As reported in the previous comment, the advantages and demerits of ocular inserts, contact, and -intraocular lens over other drug delivery systems, are added in the revised manuscript. Please see page 5 (lines 163-179) for the ocular inserts; pages (lines 383-398) for contact lens; page 17 (lines 595-608). In addition, the advantages and demerits of ocular inserts, contact, and -intraocular lens over other drug delivery systems, are resumed in the “Conclusion and Future Perspectives” paragraph (see page 21 lines 715-732).

- A figure should be self-explanatory, unfortunately only one figure is used for ocular inserts, contact, and intraocular lens with no explanation at all.

We thank the Reviewer for his/her comment. Therefore, in the revised version we have improved the figures’ description in the relative figures’ legend (see page 3 lines 61-63; page 5 lines 155-159; page 12 lines 401-405; page 18 lines 617-620).

- Introduction section must be improved. No detailed information on other conventional dosage forms used for ocular delivery is added. For instance, thiolated CDs are used as ocular permeation enhancers due to their better mucoadhesive properties.

Based on the proper Reviewer comment, we have improved the introduction section adding more information regarding other conventional dosage forms used for the ocular drug delivery (see page 3 lines 65-92)

- Conclusion is usually a summary of the abstract that highlight the major outcomes. Therefore, the conclusion should be revised in a comprehensive form.

As required, we have modified the conclusion. However, we have chosen to report the future perspectives in this paragraph which, therefore was titled “Conclusion and Future Perspectives” (see pages 21-22 lines 709-774).

Round 2

Reviewer 2 Report

The authors tried their best to address all the comments, however, some minor changes need. I mentioned in my comments to add all relevant work done in DDS for ocular delivery but the introduction still lacks some data. For instance, data related to thiolated cyclodextrin for ocular permeation is still missing.

Author Response

Reviewer 2

  • The authors tried their best to address all the comments, however, some minor changes need. I mentioned in my comments to add all relevant work done in DDS for ocular delivery but the introduction still lacks some data. For instance, data related to thiolated cyclodextrin for ocular permeation is still missing.

We thank Reviewer for his/her useful comment in response to which changes have been made. Particularly, we addedn the introduction section  the information, and the relative references, regarding the data of thiolated cyclodextrin used for ocular permeation (see page 4 lines 111-115).